# Digital Economy, Environmental Regulation and Corporate Green Technology Innovation: Evidence from China

**DOI:** 10.3390/ijerph192114084

**Published:** 2022-10-28

**Authors:** Chenggang Wang, Tiansen Liu, Yue Zhu, Meng Lin, Wenhao Chang, Xinyu Wang, Dongrong Li, He Wang, Jinsol Yoo

**Affiliations:** 1School of Economics and Business Administration, Heilongjiang University, Harbin 150080, China; 2School of Economics and Management, Harbin Engineering University, Harbin 150001, China; 3School of Management, University of St Andrews, St Andrews KY16 9XW, UK; 4Social Science Department, Heilongjiang University, Harbin 150080, China

**Keywords:** digital economy, environmental regulation, green technology innovation, mediation effect, nonlinear effect

## Abstract

**Background:** As human beings enter the digital age, the impact of the digital economy on environmental regulation and corporate green technology innovation (CGTI) is expanding. In order to effectively strengthen the efficacy of environmental regulation and improve the green technology innovation ability of corporate, this paper conducts in-depth research on the influence process of the digital economy and environmental regulation on the CGTI. **Methods:** Based on the mediating variable environmental regulation, this paper explores the influence process of the digital economy on CGTI. Combined with empirical analysis methods such as the fixed-effect model, mediating effect model, spatial model and regression analysis, the authors reveal the influence process of the digital economy on CGTI. **Results:** The digital economy can directly promote the improvement of the green technology innovation level of CGTI. The digital economy can indirectly affect the CGTI through the mediating variable of environmental regulation, marginal effect and spatial spillover effect. **Conclusions:** The digital economy and CGTI had a significant spatial correlation among different regions in China. In different regions of China, there are significant differences in the relationship between the digital economy, environmental regulation and CGTI.

## 1. Introduction

A digital economy is an economic form in which humans use digital knowledge and information to achieve high-quality economic development [1]. The digital economy can promote the rapid, optimal allocation and regeneration of resources. At the same time, the digital economy is the main economic form after the agricultural economy and the industrial economy. The digital economy develops rapidly and covers a wide range of areas. In addition, the digital economy promotes profound changes in social production methods, lifestyles and governance methods. The digital economy has also become a key force for reorganizing global factor resources, reshaping the global economic structure and changing the global competition pattern [2]. According to data released by the Ministry of Industry and Information Technology of China, the scale of China’s digital economy has ranked second in the world for several years. From 2012 to 2021, the scale of China’s digital economy has grown from CNY 11 trillion to CNY 45.5 trillion. The proportion of the digital economy in China’s GDP has increased from 21.6% to 39.8%.

Based on the development of the digital economy in the past, the digital economy will have a certain impact on the regulation of the social environment. At the same time, the digital economy will also have a certain impact on environmental regulation and corporate green technology innovation (CGTI). Among them, environmental regulation is a series of activities aimed at protecting the environment and regulating the behavior of those polluting the public environment. CGTI has developed into an important means for China to achieve “carbon peaking and carbon neutrality” [3]. However, what is the impact process of the digital economy on CGTI? What are the mediating variables in the process of the impact of the digital economy on green technology innovation? How to effectively promote CGTI capability? These issues are rarely addressed in previous studies. Although many scholars have begun to study the digital economy, CGTI, environmental regulation and other issues, there are few literatures that truly study the impact of the digital economy on CGTI from a nonlinear perspective. Therefore, in order to deeply explore the above issues, this paper deeply explores the relationship between the digital economy, environmental regulation and CGTI. At the same time, based on the conclusions of relevant scholars’ research literatures, this paper constructs relevant research hypotheses and theoretical models in the process of the digital economy’s influence on CGTI. In this paper, the threshold model, mediating effect model, spatial Dubin model and other methods are used to carry out a systematic study in combination with relevant survey data. In the course of this research, the variable relationship between the digital economy, environmental regulation and CGTI is revealed.

The research goals of this paper are as follows: (1) We hope to scientifically reveal the mechanism of the digital economy on CGTI through this research. (2) The authors hope to explore the role of environmental regulation as a mediating variable in the process of the digital economy affecting CGTI. (3) We hope to reveal the relationship between the digital economy, environmental regulation and CGTU in different regions of China. (4) We want to provide an important basis for enterprises to make decisions to enhance CGTI capacity by fully leveraging the development of the digital economy.

The follow-up structure of this paper is as follows: (1) Literature review. We sorted out relevant research literature on the digital economy, environmental regulation and CGTI. The marginal contribution of the research will also be summarized. (2) Theoretical models and assumptions. We constructed a theoretical model and proposed the research hypothesis of this paper. (3) Research design, including empirical model construction, variable design and data analysis. (4) Empirical result analysis. This part includes benchmark regression result analysis, mediation effect analysis, nonlinear effect analysis, spatial spillover effect analysis, heterogeneity analysis and robustness test. (5) Research conclusions, recommendations and limitations of the research.

## 2. Literature Review

### 2.1. The Relationship between Digital Economy and CGTI

By combing through the relevant literature of authoritative databases such as ProQuest and Emerald, we found the relationship between the digital economy and CGTI. The digital economy affects the development of CGTI from multiple perspectives. The digital economy has developed into an important part of the current Chinese economic development [4]. With the development of the digital economy, regional Internet technology, collaborative innovation capabilities, digital finance and other aspects will be significantly improved. With the development of the digital economy, the level of Internet technologies such as big data, the Internet of Things and 5G networks has been continuously improved. Therefore, high-level Internet technical support can provide an effective basic technical guarantee for CGTI [5]. The high-quality regional collaborative innovation capability can provide a strong development impetus for regional CGTI. At the same time, high-level digital finance can also provide an important financial guarantee for CGTI [6]. In addition, the digital economy can also affect the development of CGTI from the perspective of total factor integration, digital resources and industrial Internet [7].

### 2.2. The Relationship between Digital Economy and Environmental Regulation

By combing the relevant literature of authoritative databases such as SCOPUS and Web of Science, we concluded the relationship between the digital economy and environmental regulation. The digital economy helps to strengthen the efficacy of environmental regulation [8]. As the scale of the digital economy continues to expand, so does the impact of the digital economy on environmental management activities. In the process of digital economy development, the influence of data resources, new media and network environment on the social environment is also increasing. Due to the continuous emergence of new content related to the digital economy and digital resources, the content of environmental regulation closely related to the digital economy should also be constantly enriched [9]. Furthermore, with the help of the powerful functions of the digital economy such as new media and e-commerce platforms, the means of environmental regulation have also been effectively increased. Based on the efficient means of various network environments in the digital economy, the overall management efficiency of environmental regulation has also been significantly improved. It can be seen that the impact of the digital economy on environmental regulation is multi-faceted and multi-dimensional [10]. Therefore, the formulation and implementation of environmental regulations needs to take into account the relevant functions and means of the digital economy to a certain extent [11]. This will help to continuously promote the full play of the environmental regulation function.

### 2.3. The Relationship between Environmental Regulation and CGTI

By combing the literature from Emerald and Elsevier databases, we summarized the relationship between environmental regulation and CGTI. Environmental regulation will also affect the innovation activities of corporate green technology [12]. Environmental regulation can be divided into two types: command-and-control environmental regulation and market-incentive environmental regulation. Command-and-control environmental regulation mainly includes the formulation of environmental standards, pollutant discharge standards and technical standards. Market incentive-based environmental regulation mainly includes the establishment of a pollutant discharge fee, a taxation system and an emission rights trading system [13]. The effective implementation of environmental regulations can provide a favorable innovation atmosphere for CGTI. Moreover, environmental regulation can also provide a strong innovation impetus for the innovation and development of firms [14]. At the same time, in the process of implementing environmental regulations, the government will also give some firms a subsidy. Government subsidies can also promote firms to carry out CGTI activities more efficiently to a certain extent [15]. However, some scholars believe that too strict or radical environmental regulations are not conducive to the development of CGTI activities of small and medium-sized firms [16]. In addition, on the basis that CGTI has been effectively improved, the development of firms will also contribute to the effective implementation of environmental regulations [17]. It can be seen that there is a significant interactive relationship between environmental regulation and activities around CGTI.

European scholars have shown relatively abundant research on environmental regulation and CGTI. Europe is a global leader in environmental management and green technology management. European environmental policies mainly include waste management, noise pollution, chemical pollution, water pollution, air pollution, protection of the natural and ecological environment, prevention and treatment of environmental disasters [18]. Many European scholars believe that social environmental regulation is promoted along with the improvement of government environmental regulation. Therefore, the government’s environmental regulation standards must be set high. Only on the basis of high standards of both social and governmental environmental regulation can the effect of environmental regulation be fully played. For example, the European Commission has issued the *Eighth Environmental Action Plan*, the Green New Deal for Europe, the *Zero Pollution Action Plan for Air, Water and Soil* and other relevant documents on environmental protection. Among them, the requirements for environmental protection and related punitive measures are very strict [19]. From the perspective of environmental protection practice in Europe, the motivation of enterprises to carry out CGTI will increase only when the effect of environmental regulation is strong. If CGTI fails to meet the requirements of environmental regulation, the enterprise will face greater operational risks [20]. In addition, in order to maintain the long-term development of enterprises, enterprise managers will actively take measures to improve the capability of CGTI. Therefore, there is a significant interaction between environmental regulation and the CGTI activities of enterprises in Europe.

To sum up, scholars have conducted some research on the relationship between the digital economy, environmental regulation and CGTI. These researches have laid an important research foundation for the subsequent researches of scholars. However, there are still some shortcomings in the research of scholars: (1) Scholars rarely study the nonlinear relationship between the digital economy and CGTI. (2) Few scholars regard environmental regulation as the mediating variable of the impact of the digital economy on CGTI. (3) When scholars study the impact of the digital economy on CGTI, few scholars combine the spatial spillover effect to conduct research. In order to make up for the shortcomings of previous scholars in this field, this paper identifies the core issues of the study. Based on the conclusions of previous scholars, the previous research content has been further extended. This paper systematically explores the relationship between the digital economy, environmental regulation and CGTI. At the same time, the authors use a variety of methods to carry out more in-depth research, such as a mediation effect analysis, a nonlinear effect analysis, a spatial spillover effect analysis, a heterogeneity analysis and a robustness test.

The marginal contributions of this research are as follows: (1) In view of the shortcomings of previous studies on the digital economy, we study the impact process of the digital economy on CGTI. This paper enriches the research content of the digital economy. The authors also innovatively studied the nonlinear relationship between variables, enriching the research methods of digital economy-related issues. (2) Based on the research status of previous scholars, this paper further expands the research scope of the impact of the digital economy on CGTI. We introduce environmental regulation into this problem for the first time and make a thorough paper on environmental regulation as the core mediating variable. It extends the research depth of related issues. (3) This paper enriches the research methods of the impact of the digital economy on CGTI. Compared with previous studies, the authors innovatively adopt a nonlinear effect analysis, a spatial spillover effect analysis, a heterogeneity analysis, a robustness test and other research methods to solve the problems related to the digital economy. The use of these research methods reflects the marginal contribution of this research to a large extent.

## 3. Theoretical Models and Hypothesis

The development of the digital economy has accelerated the flow of social production factors and promoted the full integration of market players. The rapid development of the digital economy can directly promote the progress of CGTI [21]. At the same time, combined with the theory of environmental management, it can be seen that the digital economy could also affect the development of CGTI through environmental regulation [22]. In addition, the Internet, as the carrier of the digital economy, follows the “Metcalfe’s Law”; that is, the network value grows at the speed of the square of the number of users and the external of the network is its essence [23]. Therefore, the impact process of the digital economy may show nonlinear and spatial spillover marginal effects. Based on the above theoretical analysis, this paper will explore the relationship between the digital economy, environmental regulation and CGTI. We constructed a theoretical model and made the following assumptions (shown in Figure 1).

### 3.1. The Basic Transmission Mechanism for the Digital Economy to Influence CGTI

The impact of the digital economy on CGTI has a unique transmission mechanism: (1) The promotion of the digital economy can significantly improve the management efficiency of CGTI and environmental regulation. The digital economy provides new technologies such as big data and artificial intelligence for CGTI activities. The application of these information and digital technologies will inevitably improve the overall efficiency of CGTI to a greater extent. At the same time, affected by the development of the digital economy, the supervision mechanism for the implementation of social environmental regulations can be effectively strengthened. It will inevitably help to improve the overall management efficiency of environmental regulation [24]. (2) The digital economy helps to promote the improvement of environmental protection technology. This is mainly because the development of the digital economy has boosted the advancement of environmental management technology. Advanced environmental management technology can effectively complete the integration and efficient use of environmental management information and can also strengthen the efficient supervision of related pollution activities in society [25]. Therefore, the development of the digital economy can effectively promote the effective improvement of environmental protection technology and help to strengthen the effect of environmental protection. (3) The digital economy can provide an important guarantee for CGTI. The development of the digital economy promotes the development of the digital industry. The popularization of digital industrialization has driven the rapid development of the Internet, communications and related traditional industries. The digital development of these industries provides important support and guarantees for the overall social and economic progress. It will naturally help to optimize CGTI capability [26]. At the same time, the digital development of various industries will also be based on the mediating role of environmental regulation, which will affect the development momentum of CGTI [27]. This is mainly because the digital economy will have different impacts on both social environmental regulation and government environmental regulation, which can strengthen the comprehensive effect of environmental regulation. Based on the above analysis, this paper proposes the following research hypothesis:

**Hypothesis** **1** **(H1):**
*Environmental regulation plays a mediating role in the process of digital economy influencing CGTI.*


### 3.2. The Nonlinear Transmission Mechanism of the Digital Economy Affecting CGTI

The impact of the digital economy on CGTI is a nonlinear transmission. The digital economy based on the Internet has the characteristics of the network effect of information technology. However, with the continuous expansion of the scale of the subjects participating in the development of the digital economy, the role of the digital economy in the CGTI has become more complex [28]. Therefore, the network effects of the digital economy have become more complex. This is mainly because, with the help of many participants, the digital economy has more abundant channels for CGTI [29]. This phenomenon is also more conducive to promoting the expansion of the effect of CGTI. At the same time, with the increasing influence of the digital economy, it could provide a more efficient innovation platform, resource sharing and technical collaboration for CGTI. Therefore, the network effect characteristics are more significant [30]. In addition, at different stages of social development, the level of development of the digital economy is quite different, which leads to large differences in its efficacy [31]. Therefore, at different stages of digital economy development, the effect of the digital economy on CGTI also shows great differences. Generally speaking, as the scale of the digital economy continues to expand, the better the digital economy will promote CGTI. Based on the above analysis, this paper proposes the following hypothesis:

**Hypothesis** **(H2):**
*There is a positive nonlinear relationship between the digital economy and CGTI.*


### 3.3. The Transmission Mechanism of the Digital Economy Based on the Spatial Spillover Effect Affects the CGTI

Based on Internet information technology, the digital economy has formed a multi-category information and resource dissemination path, thus forming a relatively significant spatial spillover effect [32]. The digital economy can fully connect elements such as society, resources, economic activities and people. It could also help the digital economy to form a certain spatial spillover effect [33]. Under the influence of the spatial spillover effect, the digital economy can effectively improve the overall development of the social economy and the efficiency of resource use. For example, based on the digital economy, firms can absorb talents, funds, information and other elements from neighboring regions [34]. At the same time, firms can also use the digital economy to shorten the geographic space distance of technology alliances, improve the communication efficiency of CGTI alliances and save on communication costs [35]. Based on the above analysis, we propose the following hypotheses:

**Hypothesis** **(H3):**
*Based on the spatial spillover effect, the digital economy can effectively promote the level of CGTI in different regions.*


## 4. Research Design

### 4.1. Model

Based on the review of relevant scholars’ research combined with the theoretical model construction and research hypothesis proposed above, this research constructs the following benchmark regression model. This model mainly expresses the impact process of the digital economy on the CGTI.
(1)$$CGTI_{iy}=\alpha_{0}+cSDL_{iy}+\alpha_{2}K_{iy}+\mu_{i}+\delta_{y}+\varepsilon_{iy}$$

Among them, i indicates the society and y indicates the year. CGTI indicates the green technology innovation level of firms and SDL indicates the digital economy status of the society in which the firm is located. $K_{\mathrm{iy}}$ is the control variable, $\mu_{i}$ is the individual-fixed-effect level, $\delta_{y}$ is the time-fixed-effect condition, $\varepsilon_{\mathrm{iy}}$ is the random disturbance term and $\alpha_{0}$ is the constant term.

Based on the above analysis, in order to more directly test the role of the mediating variable of environmental regulation, we further carry out the mediation effect test. The test equation is as follows:(2)$$ER_{iy}=\beta_{0}+\alpha SDL_{iy}+\beta_{2}K_{iy}+\mu_{i}+\delta_{y}+\varepsilon_{iy}$$
(3)$$CGTI_{iy}=\gamma_{0}+c^{\prime }SDL_{iy}+bER_{iy}+\gamma_{3}K_{iy}+\mu_{i}+\delta_{i}+\varepsilon_{iy}$$

The process of the mediation effect test is as follows: (a) Regression analysis is performed by formula (1). By the regression analysis, we could test the impact process of the digital economy on the status of CGTI. (b) Based on equation (2), we examine the interrelationship between the digital economy and environmental regulation. (c) This research uses formula (3) to test the regression equation of the impact of the digital economy and environmental regulation on CGTI. If the test result is significant, it means that the mediating effect of environmental regulation is established [36].

Based on the theory of Internet development, we could see that the Internet can produce significant innovation diffusion effects. In addition, the relationship between network value and related subjects is a positive nonlinear relationship. That is to say, with the continuous expansion of network coverage, the influence of the Internet innovation diffusion effect will also increase exponentially [37]. In order to scientifically examine the complex relationship between the social digital economy and CGTI activities, this paper combines the panel threshold model to conduct a scientific analysis. The model design is as follows:(4)$$CGTI_{iy}=\varphi_{0}+\varphi_{1}SDL_{iy}\times I\left(q_{iy}\leq \theta \right)+\varphi_{2}SDL_{iy}\times I(q_{iy}\;>\theta )+\varphi_{3}K_{iy}+\mu_{i}+\varepsilon_{iy}$$

Among them, $q_{\mathrm{iy}}$ is the threshold variable, and θ is the specific threshold value; $I\left(\cdot \right)$ is the indicative function. If the variable satisfies the conditions in the parentheses, the value is 1; otherwise, it is 0. $K_{\mathrm{iy}}$ is the control variable and $\varepsilon_{\mathrm{iy}}$ indicates the random disturbance term. Equation (4) is a single-threshold model, which can be analogized to a multi-threshold model.

Finally, this paper will use the spatial Durbin model to portray the spatial spillover effects of the digital economy on CGTI. The specific space Doberman model is as follows:(5)$$\begin{array}{ll} CGTI_{iy}=\alpha_{0}+ & \rho \overset{N}{\underset{j=1}{\sum }}\omega_{ij}EGI_{iy}+\alpha_{1}\overset{N}{\underset{i\neq j}{\sum }}\omega_{ij}SDL_{iy}+\alpha_{2}SDL_{iy}+\overset{5}{\underset{k=1}{\sum }}\beta_{k}K_{iy}+\mu_{i}\\ & +\delta_{i}+\varepsilon_{iy} \end{array}$$

Among them, $\mu_{i}$ indicates the individual-fixed-effect, and $\delta_{y}$ indicates the time-fixed-effect; ρ indicates the spatial spillover coefficient, $\omega_{\mathrm{ij}}$ is the spatial weight matrix and $K_{\mathrm{iy}}$ indicates the control variable.

### 4.2. Variables

#### 4.2.1. Dependent Variable

CGTI is the dependent variable. Combining with the previous research literature of relevant scholars, the authors set the measurement index of green technology innovation as the number of scientific and technological patent applications obtained by firms [38]. The reasons for choosing the measure are as follows: (1) The level of CGTI can be accurately measured quantitatively by the number of patent applications. Moreover, the review unit of the patent application situation is generally a government agency. Therefore, the accuracy of the data is high. (2) The green technology innovation project of the firm can apply for a patent, which means that the level of the green technology innovation project is relatively high. Therefore, statistics on the number of patent applications could effectively represent the status of CGTI [39]. In the process of collecting the number of patent applications related to the CGTI, the authors mainly relies on the relevant data in the patent database of the State Intellectual Property Office of China. These data sources are authoritative.

#### 4.2.2. Independent Variable

Digital economy (SDL). In the official government statistics, there are few statistics directly related to the digital economy [40]. Therefore, this paper can only quantify and measure the development of the social digital economy based on various quantifiable indicators. Combined with the previous research literature on the digital economy by relevant scholars, the authors design a measure of the comprehensive development level of China’s social digital economy. The content of the relevant measurement indicators is shown in Table 1.

#### 4.2.3. Mediating Variable

Environmental regulation (ER). In order to scientifically measure the environmental regulation scale, this research uses the environmental index widely used by scholars as the measurement index [46]. Based on the index platform database released by Baidu, the authors analyzed 410 indexes closely related to environmental regulation and obtained the final daily average value. This indicator becomes a measure indicator to judge the status of environmental regulation.

#### 4.2.4. Control Variable

There are many factors involved in the process of CGTI [47]. In order to reduce the errors caused by missing variables, this paper introduces relevant control variables, including external dependence (OE), government intervention (GIL), socialization level (CL), factor endowment structure (ES) and science and technology investment intensity (IS). Among them, OE is measured based on the proportion of trade in GDP. GIL is measured based on the proportion of fiscal expenditure in GDP. The measurement index of CL is population density. The measurement index of ES is the ratio of capital to labor. The measurement of IS is based on the proportion of scientific research expenditure in GDP.

### 4.3. Data Sources and Descriptive Statistics

This paper selected panel data from 2011 to 2020 for research. Data sources mainly include the *China Social Statistics Yearbook*, *China Statistical Yearbook*, China Ministry of Commerce database and Baidu Index platform database. Because a few data statistics have some deficiencies, the authors use linear interpolation to complete the relevant data. In addition, in order to avoid the disadvantage of data heteroscedasticity, all data in this paper are logarithmic. Descriptive statistical results for variables are shown in Table 2.

## 5. Empirical Results and Analysis

### 5.1. Benchmark Regression Results

The regression results of the impact of the digital economy on CGTI are shown in Table 3. It can be seen from model (1) that the digital economy development coefficient is positive and significant at the level of 5%. It means that the digital economy can significantly promote the development of CGTI. In addition, model (2) adds relevant control variables on the basis of model (1). It can be seen from model (2) that the digital economy is still positive and significant at the 5% level. Therefore, hypothesis H1 is effectively verified. Furthermore, in terms of control variables, the coefficient of lnOE is negative and not significant. It means that the export-oriented economy does not significantly affect the level of CGTI. The coefficient of lnCL is positive and significant. It means that the expansion of the social scale can help to promote the sharing of social resources such as information and technology. In addition, it also helps to promote the green innovation level of local firms. The coefficient of lnGIL is positive and significant. It means that the government’s intervention in social development can help supervise or encourage the development of CGTI. The coefficient of lnIS is positive and significant. It means that the scientific research investment of firms can promote the level of CGTI. The coefficient of lnES is positive and significant. It means that firms with more capital investment are more conducive to improving their green technology innovation capabilities.

### 5.2. Mediation Effect Analysis

In addition, to verify the impact of the digital economy on CGTI, this paper will further verify the mediating role of environmental regulation. The results of the mediation effect model are shown in Table 3. It can be seen from model (3) that the impact coefficient of the digital economy on environmental regulation is significant at the level of 5%, and the coefficient is positive. As for the coefficient of the impact of the digital economy on CGTI, the coefficient in model (4) is smaller than that in model (2). It means that environmental regulation has a significant mediating effect on the impact of the digital economy on CGTI. Therefore, the regression results again verified hypothesis H1.

### 5.3. Nonlinear Threshold Effect Analysis

In order to ensure the accuracy of the research results, this paper further analyzes the impact of the digital economy on CGTI by combining the nonlinear threshold effect. Based on the panel regression model, this paper studies the development of the digital economy and environmental regulation. This paper focuses on whether the nonlinear relationship between the digital economy and CGTI changes with the changes in the digital economy and environmental regulation. First, the existence of the threshold is tested for the panel model. The inspection results are shown in Table 4. The results show that the development of the digital economy and social environmental regulation have a significant single threshold for CGTI. However, double-threshold and triple-threshold effects do not exist. Therefore, this paper could build a single-threshold regression model. The regression results of this model are shown in Table 5. The data in the table shows that when the digital economy and social environmental regulation are lower than the threshold value, the regression coefficient of the digital economy to CGTI is significantly positive at the level of 1%. It means that there is an obvious spillover effect in the digital economy. In addition, with the improvement of the digital economy and environmental regulation, when they are higher than the threshold value, the digital economy impact coefficient is also significantly positive at the level of 1%. It is higher than the previous coefficient. It means that the digital economy has a significant role in promoting CGTI. At the same time, environmental regulation has also played a positive role in regulation, showing the nonlinear characteristics of the “marginal effect” increasing. Therefore, hypothesis H2 in this paper has been verified.

### 5.4. Analysis of Spatial Spillover Effects

In order to verify the existence of spatial autocorrelation, this paper measures the Moran’s I index of the digital economy and CGTI in different regions by combining the geographical distance weight matrix. The inspection results are shown in Table 6. From 2011 to 2020, there is a significant positive spatial correlation between the digital economy and the level of CGTI. It means that there is a spatial correlation between the digital economy and CGTI in terms of spatial distribution. In the process of the SLM model test, the LR effect test only passed the time-fixed-effect test. Therefore, the time-fixed-effect model is selected in this paper. The Hausman test results show that the original hypothesis in this paper should be rejected and fixed effects should be selected. The LR test passed the significance test and rejected the original hypothesis. In other words, the SDM model cannot be simplified into the SLM model and the SEM model. The paper should be defined as a spatial SDM model with time-fixed-effects. In order to further accurately judge the spatial spillover effects of various variables on CGTI, this paper estimated various effects of digital economy development in the SDM model. Among them, the direct effect can show the impact of the social digital economy development level on the local CGTI effect. The indirect effect can indicate the impact of the digital economy of the neighboring society on the green technology innovation effect of local firms and also reflects the effect of spatial spillovers.

The dynamic SDM model regression results of time-fixed-effects and the spatial effects of independent variables are shown in Table 7. The data in the table shows that the spatial correlation coefficient of CGTI is significantly positive at the level of 5%. It means that there is a significant inter-regional interaction phenomenon in the green technology innovation activities of firms; that is, the spatial effect is significant. The CGTI in this region is positively related to the CGTI of similar firms in this region. It can be seen that the CGTI in different regions is related. The regression coefficient of the digital economy is significantly positive at the level of 1%, indicating that the development level of the digital economy could improve the level of CGTI. The regression coefficient of environmental regulation is significantly positive at the level of 5%. Thus, the higher the requirements of environmental regulation, the better CGTI will be promoted.

The above regression analysis cannot fully reflect the influence process of independent variables on the dependent variables. Therefore, this paper further decomposes the regression coefficient by partial differential and obtains the direct effect and indirect effect of each independent variable. Using these effects analysis results can more accurately analyze the interaction between variables. The specific results are shown in the last 2 columns of Table 7. The direct effect coefficient of the core variable digital economy in the model is significantly positive at the 1% level. At the same time, the indirect effect coefficient of the digital economy is significant and negative at the level of 5%. It means that the digital economy could improve CGTI in this region, but it is not conducive to the development of CGTI in neighboring regions. This is mainly due to the development of the local digital economy, which has attracted the inflow of talent, capital, information, technology and other elements from surrounding areas. This leads to the shortage of resources needed for the development of the digital economy in surrounding areas. In addition, the direct effect coefficient of environmental regulation is significantly positive at the level of 5%, while the indirect effect coefficient is not significant. It means that environmental regulation can improve the capability of firms in this region to carry out green technology innovation, but it will not affect the CGTI in neighboring regions. Therefore, the digital economy has a certain spatial spillover effect on CGTI activities. Thus, H3 is effectively verified.

### 5.5. Analysis of Regional Heterogeneity

Through the analysis of the spatial spillover effect, it is found that there is strong spatial relevance between the social digital economy, environmental regulation and activities around CGTI. Due to China’s vast territory, there are great differences in the digital economy and CGTI activities in different regions [48]. In order to further explore the relationship between the digital economy and CGTI in different regions, this paper further conducts a regional heterogeneity analysis.

Combined with relevant scholars’ research, this paper divides China into the eastern region, the central region and the western region. Based on relevant research data, the heterogeneous relationship between the digital economy, environmental regulation and CGTI in different regions is studied. The estimated results of regional heterogeneity are shown in Table 8. The data in the table shows that there are some differences in the relationship between the digital economy, environmental regulation and CGTI in different regions of China: (1) In the three regions, the impact of the digital economy on CGTI is positive. In terms of the impact of the digital economy on CGTI, the impact degree is from large to small, followed by the eastern region, the western region and the central region. (2) The impact of environmental regulation on CGTI is significantly different in significance and direction. Among them, there is a positive impact of a 5% significance level in the western and eastern regions. In the central region, there is a negative impact of a 10% significance level.

### 5.6. Robustness Test

In order to strengthen the accuracy and reliability of research conclusions, the robustness of empirical models was further tested [49]. Before the robustness test, the authors processed the data scientifically. In order to exclude the influence of extreme values, the data has been processed by tail shrinking, which is not used in the robustness test. The results are shown in Table 9. The results after the robustness test show that the coefficients of the main variables, such as the digital economy, environmental regulation and CGTI, are consistent with the previous research in terms of impact direction and significance. These conclusions show that the relevant model setting and the relevant regression results of this paper are reliable and robust.

### 5.7. Discussion

Based on the above empirical analysis, we find that there is a significant positive correlation between the development of the digital economy and the progress of CGTI. In other words, the digital economy can directly promote the level of CGTI. At the same time, the research in this paper further extends the relevant research content of previous scholars and finds that the digital economy can also indirectly improve CGTI ability through environmental regulation, which is an intermediary variable. The influence of this intermediary variable is also significant. In addition, in the process of the digital economy affecting CGTI, the digital economy further affects the development level of CGTI on the basis of marginal benefits. The impact of this marginal spillover effect is very significant. Furthermore, the digital economy can further affect the CGTI process through spatial spillover benefits. The role of this spatial spillover effect is also large. The discovery of these two “effects” also makes up for the existing deficiencies of previous scholars in this field. It shows that the impact of the digital economy on CGTI is multi-dimensional and multi-channel. Therefore, in the context of the development of the digital economy, relevant enterprises need to comprehensively consider various factors in formulating green technology innovation management measures.

## 6. Conclusions, Suggestions and Future Research Direction

### 6.1. Conclusions

The conclusions of this paper are helpful in promoting the rapid development and evolution of the digital economy in various regions of China. At the same time, the relevant suggestions put forward in this paper can provide an important reference for enterprises to make decisions related to the management and improvement of their green technology innovation ability. In addition, the proposal of relevant countermeasures in this paper also contributes to the relevant management departments of the government to make scientific management decisions and provides an important reference. Furthermore, the study conclusions of this paper can also help promote enterprises’ green technology innovation ability at a relatively novel level and also help to improve the natural environment. However, there are still some deficiencies in the research of this paper. These deficiencies need to be further improved by relevant scholars in the future.

The conclusions of this paper are as follows: (1) The digital economy can directly promote the level of CGTI. In addition, the influence process is not the result of a single factor. The digital economy factors here include information technology, information talents and digital industrialization. At the same time, the research conclusion is still reliable after the robustness test. (2) In terms of mediating effects, the digital economy can indirectly enhance CGTI capability through the mediating variable of environmental regulation. (3) There is a significant spatial correlation between the digital economy and CGTI activities in different regions of China. It means that relevant firms should not only consider their own green technology innovation activities and the development of the digital economy in the region where they are located, but also take into account the relevant conditions in surrounding regions. (4) In the eastern, western and central regions of China, there are significant differences in the relationship between the digital economy, environmental regulation and CGTI in different regions.

### 6.2. Suggestions

In order to take advantage of the opportunity period of the digital economy development to promote CGTI ability and environmental regulation efficiency, we put forward the following suggestions:

Firstly, with the help of the development of the digital economy, it provides power for CGTI. With the development of the digital economy, the application of information technologies such as big data, cloud computing, the Internet of Things, artificial intelligence and 5G communication has become increasingly mature and extensive. In this context, relevant firms can use the above information technologies to enhance their green technology innovation capabilities. Combined with cloud computing and the Internet of Things technology, firms can absorb relevant resources more efficiently and provide important basic guarantees for CGTI. Based on artificial intelligence, 5G communication and other technologies, firms can improve the speed of technological innovation and the efficiency of personnel communication in the process of CGTI. Therefore, the digital economy can effectively stimulate the promotion of CGTI.

Secondly, relevant departments should balance the level of CGTI in different regions based on the characteristics of the digital economy development. The difference in the development level of the digital economy in different regions of China leads to different impacts on firms in different regions. Therefore, in order to comprehensively promote the digital economy to promote the development of CGTI, relevant competent departments should introduce relevant management systems. On one hand, the relevant management system should be able to promote the rapid development of the digital economy in the eastern advantageous areas. Relevant management systems can improve management contents from the perspectives of the digital economy development direction guidance, regulatory mechanism, reasonable market competition and standardizing economic order. On the other hand, relevant management systems should also be able to effectively support the development of the digital economy in the central and western regions and narrow the bilateral gap. For example, they could provide important financial subsidies, tax incentives, policy guidance for the development of the digital economy in the central and western regions. Only when the development level of the digital economy in China reaches a certain level can the digital economy truly and effectively promote the improvement of CGTI capability. It is worth noting that we do not require managers in each region to develop completely different management measures to promote the local CGTI level. In areas where the development of the digital economy has limited the improvement of CGTI, we suggest that the managers in these areas should formulate targeted measures. Relevant managers should be able to actively improve the level of CGTI in these regions. These measures include increasing subsidies to relevant enterprises, increasing government investment, tax reduction and exemption, etc.

Thirdly, firms should use the efficiency of environmental regulation to optimize the effect of CGTI. In addition to the digital economy, environmental regulation also plays a key role in improving the capability of CGTI. Therefore, in the process of CGTI, firms should also fully consider the influence of environmental regulation. In particular, in the process of meeting the environmental regulation orientation, firms should develop relevant technologies that can not only meet the requirements of environmental regulation but also meet the needs of green development of firms. The environmental regulation here includes the guidance of social environmental regulation as well as the hard requirements of government environmental regulation. At the same time, firms should take environmental regulation as the driving force of CGTI. Firms can effectively promote the sustainable development of firms by developing green technologies that meet the requirements of environmental regulations.

At last, firms should fully mobilize all kinds of resources to enhance their green technology innovation capabilities. The development of CGTI activities requires the cooperation of various elements and resources, such as human resources, information, technology, R&D alliances and funds. Any lack of resources will limit the development of CGTI activities to varying degrees. Therefore, in the process of carrying out CGTI, firms should actively attract high-quality innovative talents and strengthen the ability to collect green technology-related information. In addition, relevant firms should also be able to establish green technology R&D alliances to improve the efficiency of CGTI by making use of joint efforts.

### 6.3. Research Limitations and Future Direction

Limited by the length of the paper and the time and energy of the authors, the research deficiencies of this paper are as follows: (1) The research sample of this paper still needs to be expanded. In this paper, 3091 data have been collected, all of which are from China. However, the development of the digital economy and its impact on CGTI are global. Therefore, the research sample of this paper is still relatively small. (2) The research theme needs to be improved. This paper only studied the impact of the digital economy on CGTI but did not study the impact of the digital economy on corporate organizational innovation, corporate performance improvement and corporate cost control. At the same time, due to space constraints, the impact of CGTI on the digital economy has not been studied in this paper. (3) Mediation variables need to be added. This paper only chooses environmental regulation as the mediating variable but does not choose government policy, economic environment, social innovation atmosphere or other variables as the mediating variable.

Based on the shortcomings of this paper, we put forward the following future research directions: (1) Enriching research samples. In future research, researchers can increase the research data of relevant firms outside China, especially those of some emerging economies or developed countries. The research conclusions based on the survey data of different regions will be quite different. Therefore, scholars may study different conclusions in other regions in the future. (2) Improving the research theme. In future research, scholars can further study the impact of the digital economy on corporate organizational innovation, corporate performance, corporate cost and other aspects. It will effectively enrich the research scope of related topics. In addition, scholars can also further study the impact of CGTI on the digital economy. (3) Adding mediation variables. In order to strengthen the comprehensiveness of the problem research, future scholars can add new mediating variables, such as government policies, economic environment and social innovation atmosphere.

## Figures and Tables

**Figure 1 ijerph-19-14084-f001:**
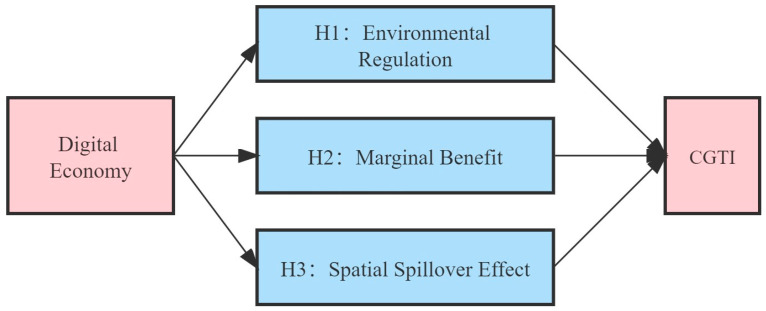
Theoretical model of this paper.

**Table 1 ijerph-19-14084-t001:** Measurement indicators of social digital economy development level.

Development level of digital economy	**Indicators**	**Data**	**Reference**
	Digital economic output	Total telecommunication services	Kostakis (2016) [41]
	Digital economy-related personnel	Number of information transmission and software practitioners	Li (2003) [42]
	Internet penetration	Broadband network access quantity per 100 people	Gupta (1997) [43]
	Mobile network penetration	Number of mobile phone users per 100 people	Conceio (2001) [44]
	Digital financial situation	Digital inclusive financial index	Soete (2000) [45]

**Table 2 ijerph-19-14084-t002:** Descriptive statistical results of variables.

Variable Category	Variable	Min.	Max.	Mean Value	Standard Deviation
Dependent variable	CGTI	0.7325	6.8713	3.1994	0.9835
Independent variable	SDL	1.2943	3.8802	2.9451	0.7381
Mediating variable	ER	0.3110	7.0426	3.9984	0.9992
Control variable	CL	1.7590	8.8842	3.9902	1.2630
	GIL	0.3382	4.2954	1.6540	0.6712
	IS	3.7108	8.9032	5.7199	1.7438
	ES	0.9325	4.5942	1.7720	1.0243
	OE	1.0843	6.7732	3.0722	0.6441

**Table 3 ijerph-19-14084-t003:** Regression results of the impact of the digital economy on CGTI.

Variable	lnCGTI	lnCGTI	lnER	lnCGTI
	(1)	(2)	(3)	(4)
LnSDL	1.831 **	1.649 **	7.621 **	1.471 ***
	(0.062)	(0.055)	(0.509)	(0.054)
lnER				0.013 **
				(0.027)
lnIS		0.431 **	0.972 *	0.390 **
		(0.027)	(0.210)	(0.047)
lnOE		0.006	0.472	0.082
		(0.017)	(0.268)	(0.061)
lnGIL		1.471 *	0.182	2.094 **
		(0.032)	(0.873)	(0.072)
lnCL		1.983 **	28.399 **	3.061 **
		(0.064)	(3.662)	(0.093)
lnES		0.390 **	7.261 ***	0.820 ***
		(0.024)	(0.833)	(0.091)
C	9.884 ***	3.092 ***	−55.771 **	−23.672 ***
	(0.142)	(0.884)	(24.083)	(3.859)
Time	Control	Control	Control	Control
Individual	Control	Control	Control	Control
N	3091	3091	3091	3091
R^2^	0.199	0.284	0.301	0.472

Note: ***, ** and * respectively indicate that the regression results are significant at 1%, 5% and 10% levels.

**Table 4 ijerph-19-14084-t004:** Analysis results of threshold existence test.

Threshold Variable	Number of Thresholds	*F*-Value	*p*-Value	BS Times	Threshold Value	1%	5%	10%
lnSDL	Single threshold	30.27	0.0000	400	-1.9722	19.0947	21.9473	17.6430
lnER	Single threshold	42.96	0.0033	400	2.4371	38.0216	33.9982	28.6438

**Table 5 ijerph-19-14084-t005:** Single-threshold regression analysis results.

Variable	Coefficient	Variable	Coefficient
Threshold value	−1.9722	Threshold value	2.4371
lnSDL × I (q ≤ −1.9722)	1.3205 ***	lnER × I (q ≤ 2.4371)	1.2884 **
lnSDL × I (q > −1.9722)	1.5719 **	lnER × I (q > 2.4371)	1.3017 **
Controlling variable	Control	Controlling variable	Control
N	3091	N	3091
R^2^	0.5291	R^2^	0.5608

Note: *** and ** respectively indicate that the regression results are significant at 1% and 5% levels.

**Table 6 ijerph-19-14084-t006:** Moran’s I index of SDL and CGTI.

Year	Moran’s I Index of SDL	Year	Moran’s I Index of CGTI
2011	0.093 ***	2011	0.014 **
2012	0.095 **	2012	0.018 ***
2013	0.099 ***	2013	0.017 **
2014	0.091 **	2014	0.022 ***
2015	0.104 ***	2015	0.027 **
2016	0.106 ***	2016	0.028 ***
2017	0.118 *	2017	0.026 *
2018	0.098 ***	2018	0.029 ***
2019	0.119 **	2019	0.033 **
2020	0.125 *	2020	0.036 **

Note: ***, ** and * respectively indicate that the regression results are significant at 1%, 5% and 10% levels.

**Table 7 ijerph-19-14084-t007:** Results of spatial model regression and spatial effect decomposition.

Variable	Main	Wx	Spatial	Variance	Direct	Indirect
lnSDL	0.288 **	−1.872 *			0.426 ***	−4.226 **
	(0.063)	(0.705)			(0.077)	(2.209)
lnER	0.045 **	−0.062 ***			0.053 **	0.063
	(0.019)	(0.033)			(0.018)	(0.014)
ρ			0.579 **			
			(0.063)			
σ^2^				0.599 **		
				(0.004)		
Control variable	Control					
Obs	3091	3091	3091	3091	3091	3091
N	310	310	310	310	310	310
R^2^	0.382	0.355	0.341	0.384	0.387	0.396

Note: ***, ** and * respectively indicate that the regression results are significant at 1%, 5% and 10% levels.

**Table 8 ijerph-19-14084-t008:** Estimation of regional heterogeneity.

Variable	Eastern Region	Central Region	Western Region
lnSDL	1.920 ***	1.884 ***	1.372 ***
	(0.126)	(0.121)	(0.106)
lnER	0.115 **	−0.084 *	0.192 **
	(0.043)	(0.017)	(0.148)
lnIS	0.392 **	0.495 **	0.158 *
	(0.050)	(0.057)	(0.033)
lnOE	0.288 **	0.142 ***	0.219 **
	(0.061)	(0.047)	(0.071)
lnGIL	1.472 ***	1.592 **	1.617 ***
	(0.306)	(0.318)	(0.339)
lnCL	1.684 **	0.044 **	2.08 **
	(0.392)	(0.117)	(0.433)
lnES	0.239 **	0.271 **	0.571 **
	(0.042)	(0.094)	(0.188)
C	7.881 *	9.685 **	12.072 **
	(2.081)	(3.702)	(5.793)
N	103	103	103
R^2^	0.719	0.703	0.754

Note: ***, ** and * respectively indicate that the regression results are significant at 1%, 5% and 10% levels.

**Table 9 ijerph-19-14084-t009:** Robustness test results.

Variable	(1)	(2)	(3)
lnSDL	2.592 **	2.638 ***	2.804 *
	(0.144)	(0.159)	(0.167)
lnER	0.019 **	0.027 *	0.022 **
	(0.014)	(0.018)	(0.014)
lnIS	0.492 **	0.571 *	0.602 **
	(0.038)	(0.041)	(0.056)
lnOE	0.134 *	0.179 **	0.281 *
	(0.020)	(0.033)	(0.049)
lnGIL	3.092 **	4.389 **	4.522 *
	(0.071)	(0.088)	(0.074)
lnCL	3.178 **	4.729 **	3.642 *
	(0.502)	(0.717)	(0.495)
lnES	0.572 *	0.690 ***	0.37 **
	(0.033)	(0.041)	(0.039)
C	8.339 **	9.041 ***	6.773 *
	(3.718)	(2.643)	(2.108)
N	392	392	392
R^2^	0.516	0.558	0.629

Note: ***, ** and * respectively indicate that the regression results are significant at 1%, 5% and 10% levels.

## Data Availability

Not applicable.

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
