# Peer review of "Digital Economy, Environmental Regulation and Corporate Green Technology Innovation: Evidence from China"

_ijerph, 2022, doi:10.3390/ijerph192114084_

Round 1

Reviewer 1 Report

Review

Of the article "Digital Economy, Environmental Regulation and Innovation in Green Corporate Technologies: Evidence from China"

 Methodological notes:

Abstract does not meet the basic requirements related to signaling the structure of the study.

Its structure should be clearly built according to the pattern:

PURPOSE: Make clear what the main goal and specific goals of the article are.

METHODOLOGY: The research problem should be formulated based on the analysis of the literature on the issues of digital economy and environmental regulations and innovations in the field of green corporate technologies. 

 FINDINGS: The analysis of the digital economy, and environmental regulations and innovations in the field of green corporate technologies, allowed to establish: the findings should be indicated.

IMPLICATIONS: - for the economy, - for local communities, - for stakeholders, - for the natural environment. Determine what are the implications for science and further research 

ORIGINALITY AND VALUE: Identify the relationship of the digital economy and its impact on environmental regulation and innovation in corporate green technologies

Methodological notes:

In the introduction, the justification for the choice of the topic of the work should be indicated, based on the scientific literature, especially the literature that is related to the aim of the work and research hypotheses. The proposed literature is one-sided, mainly by authors dealing mainly with the issues of digital economy, innovation and technology. Studies related to environmental regulations and green technologies are cited to a small extent. It is necessary to indicate examples of environmental regulations and green technologies from outside China, e.g. from the European Union (EU). Today, the EU is a global leader in innovation in the field of environmental regulations and green technologies. The authors do not provide examples from the EU area and do not cite the authors of the studies from the EU countries.

Lack of description and justification of the proposed research problem. The justification of the research problem should be based on the indication of the main goal and specific goals, including: methodological goal, application goal.

The formulated hypotheses, their description and the attempt to prove it is very problematic, because it is not based on statements resulting from the quantitative and qualitative research carried out, on the models used, but on assumptions and possible probabilities.

No analysis or evaluation of the conceptual differentiation of the proposed research issues has been performed. The conceptual analysis should be based on a review of the scientific literature.

The process of selecting the basic literature relating to the research objectives was not indicated. Failure to indicate the sources of publications, eg ProQuest, Emerald, SCOPUS, weakens the scientific quality of the study.

The proposed article does not indicate the methods and techniques of collecting, processing and analyzing secondary data (desk research), eg CAWI, IDI, ITI, FGI.

The above remarks are important from the point of view of the scientific level presented in the article, which means that the article has local significance.

Author Response

Dear Ms. Gabrielle Yang and reviewers,

We are very grateful to your constructive suggestions that really help a lot to improve this study. In our revised manuscript, we considered all of your suggestions and then did a careful revision. Almost all of texts have been improved, including the literature, data, discussion and research goals. We marked the red on the major revised contents. Following, we introduce how we improved this study following your guidance.

Reviewer 2 Report

The article deals with a current and important issue. However, it needs some improvement. Firstly, the "discussion" section is missing, which should contain explicit references to the research results published by other authors (it should be shown here to what extent new research confirms, extends, supplements, contradicts the results of previous research and what are the changes in our understanding of a given phenomenon) . Questions also arise in the reader's mind: Is the growth of the digital economy itself not also driven by CGTI innovations as independent variable (e.g. as part of a feedback mechanism)? Could be the number of patent applications from each region a comparable CGTI indicator? Is it really necessary to develop different regulatory and legal solutions in individual regions to increase the potential of CGTI (because e.g. in the European Union, unification of regulations is used for this purpose in order to create the phenomenon of convergence, "equation to the best")? These questions should be also addressed in the paper.

Author Response

(The authors gave the same response as above.)
